# Load Estimation in Onshore Wind Farms Using Surrogate Modeling and Generic Turbine Models

Alexander Mönnig<sup>1,2</sup>, Ansgar Hahn<sup>1</sup>, Astrid Lampert<sup>3</sup>, and Ulrich Römer<sup>2</sup>

<sup>1</sup>Alterric GmbH, Holzweg 87, 26605 Aurich, Germany

<sup>2</sup>TU Braunschweig, Institute for Acoustics and Dynamics, Langer Kamp 19, 38106 Braunschweig, Germany

<sup>3</sup>TU Braunschweig, Institute of Flight Guidance, Hermann-Blenk-Str. 27, 38108 Braunschweig, Germany

Correspondence: Alexander Mönnig (alexander.moennig@alterric.com)

Abstract. This article investigates the development and application of surrogate models, based on slightly adapted generic turbine models, for predicting loads on real-world wind turbines. A small set of aeroelastic simulations provided training data for both Polynomial Chaos expansion and Gaussian Process regression models, which were trained to predict blade loads, tower accelerations, and their respective seed-to-seed variability. To evaluate the practical suitability of these models a case

study was performed. Here, the surrogate models were applied to predict blade loads and tower accelerations respectively, using five years of SCADA data from an onshore wind farm. While the models approximated the real-world turbine behavior with a reasonable accuracy, the prediction quality varied across the different turbines in the park and was further influenced by factors such as the turbine's operational years and diurnal patterns suggesting a correlation with the turbulence intensity. Despite some limitations, the findings support the practicality of developing surrogate models for enabling efficient load estimations.

# 10 1 Introduction

Reducing downtimes and extending the lifetime of wind turbines are two important means for improving the profitability and hence, for increasing the future share of wind energy. These goals could be reached by assessing historic and managing future turbine loads based on aeroelastic simulations, which in turn could enable the optimization of operation and maintenance (O&M) strategies.

- 15 Two factors can be indentified, which limit the large scale application of aeroelastic simuators to these specific use cases:
  - 1. **Computational burden:** In most use cases, a large number of evaluations are required to cover the range of operating conditions and obtain results that are statistically meaningful, where the latter problem originates in the inherent stochastic nature of turbulence modeling and an associated seed-to-seed variability in the simulations.
  - 2. Limited access to accurate turbine models: High-resolution turbine models are usually proprietary to turbine manu-
- 20

facturers. This particularly limits the application of aeroelastic simulators in operational settings where load assessments could benefit turbine operators to improve their O&M strategies.

To address the computational burden, surrogate models have emerged as a promising solution to reduce the computational cost, while still providing accurate representations of complex simulation frameworks.


Various types of surrogate models, including the Polynomial Chaos Expansion (PCE), Murcia et al. (2018), Artificial Neural Networks (ANN), Gasparis et al. (2020), Response Surface Methods (RSM), Toft et al., and Gaussian Process Regression (GPR), Wilkie and Galasso (2021), have shown significant potential in accelerating the estimation of aeroelastic loads from wind turbine simulators.

Previous research has thereby explored surrogate models with different purposes. For example, Murcia et al. (2018) employed PCE to predict fatigue loads and power generation for isolated wind turbines under site-specific inflow conditions. In contrast,

Dimitrov (2019) developed surrogate models capable of predicting loads within a farm layout, incorporating parameters such as row spacing and the number of upstream turbines.

While surrogate models offer significant computational advantages in their execution, the stochastic nature of aeroelastic simulation due to turbulence-related seed-to-seed variability can make their development still cost intensive. In the literature, two main approaches have been used to handle this variability. The first class of methods requires realizations with different seeds

- 35 using the same vector of operational parameters. While only six realizations are recommended for estimating the mean of the loads by the IEC (International Electrotechnical Commission) 61400-1:2019, capturing the variance across these realizations requires a significantly larger sample size. For instance, Murcia et al. (2018) utilized 100 realizations in their study. Because of the high cost associated with to the seed-to-seed variability, authors often focus solely on the mean prediction with a limited set of realizations. For instance, Dimitrov et al. (2018) used eight replications per sample while focusing primarily on the mean
- 40 loads. On the other hand, Murcia et al. (2018) employed two PCE models to predict both the mean and standard deviation of loads, capturing the heteroscedastic nature of load variance, though at a high computational cost. In a more general context, a surrogate model based on the generalized Lambda distribution and regression data with replication has been proposed in Zhu and Sudret (2020).

The other category of approaches, instead, does not rely on replications. GPR is to be mentioned here, as a classical example.

More recently, replication-free versions of the generalized Lambda surrogate model, Zhu and Sudret (2021), and random field based surrogates, Lüthen et al. (2023), have been proposed.

To address the second factor - limited access to accurate turbine models - Reference Wind Turbine (RWT) models could provide a potential workaround. These pre-implemented generic turbine models, cf. Gambier (2022), are openly available and have been used extensively in research. While RWT models are not exact representations of specific commercial turbines, they

capture the fundamental physics and design principles of modern wind turbines. This could make them potentially suitable for preliminary load assessments and as a starting point for model refinements to better match specific turbine characteristics. In this article we investigate whether RWT-models, in combination with a surrogate-based simulation of turbine loads,

can overcome the challenges of computational burden and limited access to accurate turbine models identified earlier. The study builds on a large database of historic SCADA data that can be used to assess the accuracy of the combined surrogate -

RWT model. Specifically, we will analyze the viability of the generated surrogates in a case study by applying them for the prediction of selected blade bending moments and tower vibrations recorded in an onshore wind farm in Germany. Given that the turbulence-induced variance sets a natural limit on predictive accuracy, our surrogate models - based on PCE and GPR - will estimate both the mean and the standard deviation of the loads.


The remainder of this paper is structured as follows: Section 2 presents the methodology. Section 3 outlines the case study 60 setup which serves to evaluate the proposed approach. Section 4 presents the results of the model training and application. Finally, some conclusions are drawn and directions for future research are suggested.

# 2 Surrogate Modeling

Complex numerical models cannot be used directly for carrying out parametric studies, uncertainty quantification and other multi-query tasks. It is therefore common to employ surrogate models to reduce the computational complexity while simultaneously controlling the approximation error. This section will present theory and methods employed for constructing the surrogate models in this study. The process of developing the surrogate models is here divided into four key steps:

- 1. Identification of features for characterizing load variations.
- 2. Characterization of site-specific operating conditions through appropriate distributions and sample generation.
- 3. Creation of a load database linking operational conditions to specific turbine loads via aeroelastic simulations.
- 70 4. Construction of the surrogate models.

These steps are detailed in the following sections, where we present our modeling choices. An implementation is then presented in Section 3, where surrogates are built for reconstructing historic loads of an onshore wind farm over a time period of 5 years.

### 2.1 Features for Characterizing Load Variations

- The surrogate models are developed using four features identified from similar load studies, see e.g. Dimitrov et al. (2018). 75 Note that, unlike some recent work by Dimitrov (2019), we exclude wind farm layout features and instead use the turbulence intensity of each (wake-affected) inflow to account for farm effects. As this study focuses on reconstructing loads from 10minute SCADA data, we selected features that are available (directly or indirectly) from industry-standard SCADA systems. The features considered here are as follows and always refer to the 10-minute average values:
- **Horizontal wind speed**  $V_{\text{mean}}$  [m/s]: It is directly available in typical SCADA systems. 80

•

Air density  $\rho_{\text{mean}}$  [kg/m<sup>3</sup>]: While the air density is seldom directly measured in a turbine, an approximation is possible using the ideal gas law as

$$\rho_{\text{mean}} = \frac{p}{RT_{\text{mean}}} = 3.4837 \frac{p}{T_{\text{mean}}} \tag{1}$$

. . . . . . . .


54. 4.

with the pressure 
$$p$$
, the average air temperature  $T_{\text{mean}}$  measured at the hub height and the general gas constant  $R = 8.314 \text{ J} \text{ mol}^{-1} \text{ K}^{-1}$ . The pressure  $p$  at hub height can be approximated using

$$p = 101.29 - (0.011837)z + (4.793 \times 10^{-7})z^2,$$
<sup>(2)</sup>

where z is the hub height in meters, see Manwell et al. (2009, Chap. 2). This is a simplified approach that does not capture any weather-related variations in the air pressure, for instance due to high or low pressure systems, temperature inversion or different moisture content.

**Turbulence intensity** I [-]: The turbulence intensity I can be directly estimated from the available SCADA measurements of wind, utilizing 10-minute SCADA statistics as

$$I = \frac{V_{\rm std}}{V_{\rm mean}},\tag{3}$$



where  $V_{\text{std}}$  represents the standard deviation and  $V_{\text{mean}}$  the average of the wind speed in any given 10-minute period. However, the usual anemometer placement behind the rotor distorts these measurements. While the 10-minute average wind speed is commonly internally adjusted to account for the deceleration across the rotor plane, such corrections are not applied to the standard deviation of wind speed. We therefore adopt the method implemented by Barthelmie et al. and Jørgensen et al. (2003) to approximate  $V_{\text{std}}$  for below-rated conditions as

$$V_{\rm std} = \frac{P_{\rm std}}{\left(\frac{dP}{dV}\right)_v \cdot B},\tag{4}$$

where  $P_{\text{std}}$  is power's standard deviation,  $\left(\frac{dP}{dV}\right)_v$  is the power derivative with respect to wind speed, and B is an empirical correction factor. For above-rated conditions, we extend the method to use the pitch angle  $\beta_{\text{std}}$  instead of the power as

$$V_{\rm std} = \frac{\beta_{\rm std}}{\left(\frac{d\beta}{dV}\right)_v \cdot B}.$$
(5)

To obtain the necessary power and pitch curves, data from Original Equipment Manufacturers (OEM) or regression models applied to historical operational data may be used.

Yaw misalignment angle  $\psi_{\text{mean}}$  [°]: It is directly available in typical SCADA systems.

# 105 2.2 Sampling from Site-Specific Distributions

Depending on the scope of the surrogate (i.e., single wind farm or wider region), distributions of the turbines' operating conditions have to be derived, so that training and validation samples may be drawn from the resulting joint distribution. Table 1 presents an example of how these distributions could be defined. For the wind speed, a Beta distribution could be employed, as suggested by Dimitrov et al. (2018), with the aim of generating more samples at low wind speeds where the variance of the

remaining parameters is usually larger. The shape parameters  $\alpha$  and  $\beta$  may be found empirically through iterative adjustments until the desired shape is achieved and the minimum (min<sub>V</sub>) and maximum wind speed (max<sub>V</sub>) would align with the cut-in and cut-out wind speed of the considered turbines.

Here, the turbulence intensity, air density, and the yaw misalignment angles are modeled using distributions that solely depend on the wind speed. The bounds for these distributions can be either established using physical laws, as shown in Dimitrov

et al. (2018), or derived directly from available historic measurements, for example, through an interpolation approach. In the


Table 1. Probability distributions for the chosen feature variables.

| Parameter                          | Distribution                                                                                         |
|------------------------------------|------------------------------------------------------------------------------------------------------|
| Wind Speed $(V_{mean})$            | $\text{Beta}(\alpha,\beta,\min_{V_{\text{mean}}},\max_{V_{\text{mean}}})$                            |
| Air Density ( $\rho_{mean}$ )      | $\text{Uniform}(\min_{\rho}(V_{\text{mean}}), \max_{\rho}(V_{\text{mean}}))$                         |
| Turbulence Intensity $(I)$         | $Uniform(min_I(V_{mean}), max_I(V_{mean}))$                                                          |
| Yaw Misalignment ( $\psi_{mean}$ ) | $\operatorname{Uniform}(\min_{\psi}(V_{\operatorname{mean}}), \max_{\psi}(V_{\operatorname{mean}}))$ |

example of Table 1, uniform distributions were selected, for simplicity.

Once the distributions are defined, training samples can be drawn from the input space. A common approach is to use the Sobol sequence to generate samples within a 4D-hypercube, where each dimension represents a uniformly distributed random variable. Subsequently, the samples must be transformed to the correlated distributions defined in the previous section. As we deal with a set of dependent variables, the inverse Rosenblatt transformation is a suitable function for this task, Dimitrov et al. (2018). Assuming just two independent variables  $u_1$  and  $u_2$  for simplicity, it is defined as

$$\begin{cases} \theta_1 = F_{\theta_1}^{-1}(u_1), \\ \theta_2 = F_{\theta_2|\theta_1}^{-1}(u_2|u_1), \end{cases}$$
(6)

see Mara and Becker (2021). In this example,  $F_{\theta_1}^{-1}$  represents the inverse Cumulative Distribution Function (CDF) of  $\theta_1$ , and 125  $F_{\theta_2|\theta_1}^{-1}$  is the inverse CDF of  $\theta_2$  conditioned on  $\theta_1$ .

In addition to the training samples, a set of validation samples is required for measuring model performance. These samples can be drawn randomly.

# 2.3 Generating the Load Database

The next step involves establishing a load database that links each sampled operating condition to the corresponding turbine 130 loads through an aeroelastic simulator f as

$$y = f(\boldsymbol{\theta}, \omega), \tag{7}$$

where  $\theta$  represents the vector of input features (wind speed, air density, turbulence intensity, yaw misalignment) and y any given target variable (e.g., blade root bending moments). Moreover,  $\omega$  represents an elementary random outcome to account for the seed-to-seed variability. Please note that temporal averages and standard deviations are denoted with mean, std, whereas

mean and standard deviation with respect to randomness ( $\omega$ ) are denoted with  $\mu$ ,  $\sigma$ , respectively. These simulations require high-resolution turbine models, often unavailable to operators. This study explores the option of using open-source RWTs to approximate loads on real-world turbines. RWTs cover various turbine types across different power classes, rotor diameters, and hub heights, see Gambier (2022). While they cannot perfectly match commercial turbines, they may serve as a good

starting point for further adaptations. A workflow for such an approach could look, as suggested by Barter in the NREL 140 forum<sup>1</sup>, as follows:

- 1. Select an appropriate RWT model which best approximates the real-world wind turbine.
- 2. Adjust key parameters (e.g. tower height, rotor diameter, rated power).
- 3. Reoptimize and recalculate the modified structures using WISDEM (Wind-plant Integrated System Design and Engineering Model), see Dykes et al. (2021).
- 145 4. Reture the controller using ROSCO (Reference OpenSource Controller), see NREL (2021).
  - 5. Generate new OpenFAST (Fatigue, Aerodynamics, Structures, and Turbulence) input files via WEIS (Wind Energy with Integrated Servo-control), see Abbas et al. (2022).
  - 6. Simulate the loads (e.g. blade bending moments) for each sampled operating condition using OpenFAST, see Buhl et al. (2023).
- 150 Therefore, emulating an OEMs design process in a simplified manner should allow turbine operators to obtain turbine models sufficient for initial load assessments on real wind turbines.

# 2.4 Surrogate Methods

With the generated training and validation data, regression models can be applied to capture the input-output relationships. The case study presented in Section 3 evaluates PCE and GPR. These techniques were selected for their proven ability in capturing
the load response of wind turbines (Dimitrov et al. (2018); Murcia et al. (2018); Slot et al.; Teixeira et al.; Gasparis et al. (2020)) also with a moderate size of the training data. While additional approaches were investigated - including stochastic polynomial chaos expansion (SPCE) in a follow-up study - these methods did not yield significant improvements over the approaches presented here. The following sections provide a brief theoretical background for both PCE and GPR.

# 2.4.1 Polynomial Chaos Expansion

The PCE is a methodology aims at approximating models  $y = f(\theta, \omega)$  featuring a finite number of random input variables M summarized in the vector  $\theta = (\theta_1, \dots, \theta_M)$ . For  $\omega$  fixed, the model response y is considered to be a random variable with finite variance. Hence, a set of basis polynomials over  $\theta$  can be found for its effective approximation. The polynomials must thereby be orthogonal with respect to the distribution of each input variable, see Xiu and Karniadakis (2002). Once such a set

<sup>&</sup>lt;sup>1</sup>Garret Barter in Reference Turbines for Scaling - Wind & Water, https://forums.nrel.gov/t/reference-turbines-for-scaling/3601/4, 2022

of polynomials is identified, the moments of y may be expanded as

$$\mu(\boldsymbol{\theta}) = \sum_{j=0}^{\infty} c_j \phi_j(\boldsymbol{\theta}), \qquad (8)$$
$$\sigma(\boldsymbol{\theta}) = \sum_{j=0}^{\infty} d_j \phi_j(\boldsymbol{\theta}), \qquad (9)$$

where  $\mu(\theta), \sigma(\theta)$  denote the input-dependent mean value and standard deviation of y, averaged over  $\omega$ . We focus on the second order moments because of the limited sample size available. Moreover,  $\phi_j(\theta)$  is a multidimensional polynomial basis function

$$\phi_j(\boldsymbol{\theta}) = \phi_{j_1}(\theta_1) \cdots \phi_{j_M}(\theta_M) .$$
(10)

The  $c_j, d_j$  are the polynomial coefficients. Moreover, j is the enumeration of a multidimensional index element, i.e.,  $\mathbf{j} = (j_1, \dots, j_M)$  with  $\mathbf{j} \leftrightarrow j \in \mathbb{N}$ , that denotes the degree of the polynomial in each dimension, see Murcia et al. (2018). In practice, this expansion is truncated to a finite number of  $N_c$  terms, for instance the mean value approximation reads

$$\mu(\boldsymbol{\theta}) \approx \sum_{j=0}^{N_c-1} c_j \phi_j(\boldsymbol{\theta}) \,. \tag{11}$$


One way of finding the  $N_c$  values of the coefficients is through the use of regression methods, e.g. with the least-squares or the Least Absolute Shrinkage and Selection Operator (LASSO) approach, as proposed by Tibshirani (1996).

The method as described so far usually assumes that the M input variables are uncorrelated. In cases where this assumption does not hold, the input space should be first transformed into an uncorrelated one using a suitable transformation function (e.g. the Rosenblatt function). Alternatively, a dedicated PCE for dependent variables can be used, see Jakeman et al. (2019).

# 180 2.4.2 Gaussian Process Regression

GPR is a non-parametric approach where, in our context, the functions  $\mu(\theta), \sigma(\theta)$  are assumed to be drawn from a Gaussian process. Each process is defined by a mean and covariance function. While the mean is often assumed to be zero, a common covariance function for two points  $\theta$  and  $\theta_*$  is the Radial Basis Function (RBF) kernel

$$\mathcal{K}(\boldsymbol{\theta}, \boldsymbol{\theta}_*) = \sigma_{\rm f}^2 \exp\left(-\frac{\|\boldsymbol{\theta}_i - \boldsymbol{\theta}_*\|^2}{2\ell^2}\right).$$
(12)

Here  $\ell$  corresponds to the length scale of the kernel, i.e. the distance for which random variables are correlated. The standard deviation of the GP is controlled with  $\sigma_{f}$ . Assuming a vanishing mean for simplicity, the training and test samples of are jointly distributed as

$$\begin{pmatrix} \boldsymbol{\mu}_{\boldsymbol{\Theta}} \\ \boldsymbol{\mu}_{*} \end{pmatrix} \sim \mathcal{N} \left( \mathbf{0}, \begin{bmatrix} \mathbf{K}_{\boldsymbol{\Theta},\boldsymbol{\Theta}} + \sigma_{n}^{2} \mathbf{I} & \mathbf{K}_{\boldsymbol{\Theta},*} \\ \mathbf{K}_{\boldsymbol{\Theta},*}^{\top} & \mathbf{K}_{*,*} \end{bmatrix} \right),$$
(13)


where we focus on the GP approximation of  $\mu$  for simplicity. Through pairwise evaluation of the covariance function  $\mathcal{K}$ , the covariance matrices between the training points ( $\mathbf{K}_{\Theta,\Theta}$ ), between the test points ( $\mathbf{K}_{*,*}$ ) and between the training and test points ( $\mathbf{K}_{\Theta,*}$  and  $\mathbf{K}_{*,\Theta}$ ) are computed. The term  $\sigma_n^2 \mathbf{I}$  models the noise in the observed samples  $\mathcal{N}(0,\sigma_n^2)$ , where  $\mathbf{I}$  is the identity matrix.

The posterior distribution is obtained by conditioning the joint Gaussian on the training data  $\mathcal{D} = \{\Theta, f_{\Theta}\}$ , yielding the predictive mean and covariance

 $\mathbb{E}[\boldsymbol{\mu}_*|\mathcal{D}] = \mathbf{K}_{*,\boldsymbol{\Theta}} [\mathbf{K}_{\boldsymbol{\Theta},\boldsymbol{\Theta}} + \sigma_n^2 \mathbf{I}]^{-1} \mathbf{f}_{\boldsymbol{\Theta}},$   $\operatorname{Cov}[\boldsymbol{\mu}_*|\mathcal{D}] = \mathbf{K}_{*,*} - \mathbf{K}_{*,\boldsymbol{\Theta}} [\mathbf{K}_{\boldsymbol{\Theta},\boldsymbol{\Theta}} + \sigma_n^2 \mathbf{I}]^{-1} \mathbf{K}_{\boldsymbol{\Theta},*}.$ (14)

# 3 Case Study

To evaluate the effectiveness of generic turbine models in predicting real turbine loads, the previously discussed methodology is applied in a case study. This study focuses on a wind farm in northern Germany operated by Alterric. The farm consists of eight turbines in the 3 MW range. Further specifications cannot be disclosed in this study. Historical SCADA data at 10-minute intervals are available for each turbine from 2017 to 2022. Figure 1 illustrates the farm's layout and the mean wind conditions deriv