# Peer review of "Load Estimation in Onshore Wind Farms Using Surrogate Modeling and Generic Turbine Models"

_Wind Energy Science, 2025_

## Referee Comment (RC1)

**General Comments**

This paper investigates the use of generic turbine models combined with surrogate models to predict loads on real-world turbines, validated through comparison with multi-year measurements from an operational wind farm. The motivation of the paper is highly relevant, addressing a topic of growing interest in the wind energy community. The manuscript is overall well structured and generally well written. However, the current performance of the surrogate models does not appear sufficient to draw robust conclusions about the method's applicability. I recommend the authors either reframe the manuscript as a methodological paper or extend the work on the surrogate models to improve their predictive capabilities. Specific suggestions are provided below.

**General Writing**

Throughout the paper, please integrate references more smoothly into the text rather than relying on "see" or "cf.".

**Abstract**

It is a great short and direct abstract. However, it would benefit from more emphasis on the key challenge the authors are trying to tackle here: the lack of access to commercial turbine models and whether generic models can address high-level design considerations.

**Introduction**

- The introduction is overall good. For clarity, please address the limited access to accurate turbine models first (before the surrogate modelling), as this is the core novelty of the work.
- Please include a brief literature review on the use of generic turbine models in other studies.

**Section 2: Surrogate Modelling**

- Line 100: Please elaborate on the methodological extension related to the pitch angle.
- Line 121: The reference to Dimitrov et al. (2018) is abrupt. Adding more context would make it smoother.
- Line 124: the reference to Mara and Becker should also be introduced with more context.
- Line 139: The introduction of the forum is somewhat weird. Maybe the best way to do it is to remove this reference and explain this as the procedure you will follow with some justifications. It might also improve clarity to reorganize this as a dedicated section titled "Adapting the Generic Turbine Models," providing a more detailed explanation of each stage of the parameters updates. This section could be just after the introduction.

- Line 150: Please clarify what error thresholds or maximum acceptable errors define the model applicability.
- Line 152: Mentioning SPCE as a future research direction is a bit early in the paper. Consider moving this to the discussion or conclusion, elaborating on its potential. References to other approaches should either be briefly presented with names and results at some point in this paper or not mentioned at all.

**Section 3: Case Study**

- Line 205: Please clarify the limits under which generic turbine models can be considered applicable to real-world scenarios.
- Line 224: citations to be improved.
- Section 3.2: The validation of the generic model is lacking. Before introducing surrogate models, a comparison between generic turbine loads and measurements under different DLCs is essential. If this has been done, please include at least key figures, if not, it represents a crucial missing step.
- Figure 2: This figure is currently not cited in the paper. It would improve the methodology section to reference it when discussing the joint distribution and bounds definition.
- Line 258: Indicate the OpenFAST version used.
- Line 264: Justify the use of 30 seconds as transient time with relevant references.
- Line 277: Explain the motivation for using the Rosenblatt transformation with GPR. Have you tuned the hyperparameters (using grid search, random search, or Bayesian optimization)? This is expected to improve model performance.

**Section 4: Results**

- Line 290: The opening sentence is unclear. The sensitivity analysis on polynomial order and training set size you mentioned should be first introduced and then presented, including the maximum training set size used.
- Line 300: The current surrogate models seem not sufficiently capture dynamic and nonlinear effects, you should here refer to the other models you tried, and the database should be extended, 300 samples is very little for 4 input parameters.
- Line 304: Clarify what the "respective other model" refers to.
- Line 307: Specify which convergence analysis is being referenced.
- Line 310: Provide the mentioned error curves in the manuscript.
- Figure 3: The figure is difficult to interpret. Consider restructuring it, good examples could be Figure 3 and 4 in https://doi.org/10.5194/wes-9-1885-2024, showing training sample positions relative to measurements, followed by load predictions relative to inputs. In addition to that, it can be seen on the figure that many training points do not align with measurements, potentially explaining low model performance.

In Section 4.1, the surrogate models predict mean values adequately (except for maximum tower-top acceleration), but their variance predictions are off. Before moving on to further analysis, the following points need to be studied:

- Quantify the error between generic model outputs under measured environmental conditions and actual measurements.
- Increase the training set size, 300 samples with four input variables is small. Please show the convergence analysis mentioned earlier and consider extending to 1000 samples for instance.
- Perform a convergence study on the number of seeds to take into account to capture the seed-to-seed uncertainty.
- Discuss other surrogate models, such as neural networks, which have shown promising results in similar contexts.
- Address the poor performance at low wind speeds, and elaborate on way to address it, expanding the dataset could be a way.
- Tune the hyperparameters of the models using grid search or random search for instance.

**Section 5: Discussion**

- Please revise conclusions to reflect that surrogate models captured mean values adequately but struggled with variance and extreme events.
- Line 376: The expectation that increasing the number of turbulent seeds will resolve discrepancies lacks evidence. Please conduct and present a convergence analysis on maximum tower acceleration with respect to turbulent seed number and training set size.

**Summary and Recommendations**

The scope of work is highly relevant, and the preliminary findings on turbine aging and inter-turbine variability are promising. However, substantial improvements are required before publication. Key recommendations are:

1. Provide an error estimation from discrepancies between the generic OpenFAST model and measurements.
2. See the impact of increasing the size of the training dataset on the model performances.
3. Elaborate on surrogate model hyperparameter tuning, considering grid search or random search.
4. Expand the discussion on alternative models tested, consider implementing neural networks with larger datasets.
5. Add sensitivity analyses on the number of turbulent seeds and the size of the training/validation sets.

---

## Referee Comment (RC2)

**General comments:**

In this study, the authors developed and trained two surrogate models for predicting wind turbine loads and evaluated their performance using SCADA data. The surrogate models exhibited varying levels of accuracy in load prediction. The topic of using surrogate models for turbine performance evaluation is current and relevant to ongoing research in the field. The paper is generally well-structured, with a logical and coherent flow. However, it does not clearly emphasize its innovations, and the interpretation of the results is not fully convincing. The methodology, including the use of PCE and GPR models and reference wind turbine model, is well-established in the literature, and thus do not present novelty to the reader.

Furthermore, the interpretation of the prediction accuracy is questionable. For example, the annual median error in tower acceleration exceeds 40%, which raises concerns not only about the accuracy of the surrogate model itself, but also about the consistency between the reference turbine used in the OpenFAST simulation and the actual on-site load measurements. This concern hinders the reproduction, generalization, and practical application of the proposed approach (surrogate + generic WT models).

The following comments are provided for each section of the manuscript, with the hope of improving the quality of the paper.

**Abstract:**

 The authors present a concise abstract that briefly introduces the methodologies applied in the study, and the conclusion regarding the use of surrogate models for load prediction. However, the research question is not clearly stated, leaving the reader uncertain about the specific problem the study aims to address. In addition, the claim of achieving 'reasonable accuracy' (line 7) requires clarification, as the blade load and tower acceleration report distinguish errors. This variation should be acknowledged and discussed more explicitly.

**Introduction:**

In this section, the authors identify two key challenges in aerodynamic simulations: high computational cost and limited access to detailed turbine models. Surrogate models are then proposed as a means to reduce computational effort, while the adoption of a reference wind turbine model is suggested as a potential solution to the issue of limited public model.

However, the motivation or assumptions the selection of PCE and GPR models require further clarification. In addition, the literature review does not identify existing gaps in the application of surrogate models, particularly PCE or GPR, for wind turbine load prediction. As a result, the novelty or contribution of the study is not convincingly demonstrated.

Although the authors aim to address data limitations by using a reference turbine model, the lack of either public available real turbine models or SCADA datasets makes it difficult to assess the broader significance of this study or novelty of the approach.

1) The authors state that the development of surrogate models remains cost-intensive due to the randomness associated with wind turbulence. However, the discussion around the number of

seeds used in simulations is unclear. While the cited literature uses varying seed numbers (e.g., 6, 100, 8), the paper does not provide a clear conclusion on what constitutes a sufficient number of seeds for accurate load prediction. It would be valuable to clarify how the number of seeds affects the reliability or prediction accuracy of the surrogate model, and what the potential implications are of using an insufficient number, such as the recommended value of 6 by IEC standard.

While the PCE model used in Murica et al. (2018) required 100 realizations to predict the mean and standard deviation of the load, the authors of this study describe that approach as having a "high computational cost" (line 40). However, this study also adopts the PCE model and claims it "can overcome the computational burden" (line 40), without providing a clear explanation of the number of seeds used or the rationale for selecting the PCE model over other alternatives such as ANN, RSM, or GPR, which are mentioned briefly in lines 25–26. A more detailed justification for the model choice and a discussion of its computational trade-offs would strengthen the study, and reduce potential confusion for readers.

In line 44, the GPR model, described as not relying on replication, is also selected in this study, which helps to overcome the high computational burden associated with turbulence-induced randomness. However, the differences between the PCE and GPR models are not clearly presented, making the rationale behind the selection and comparison of these two models unclear to the reader. Especially, the GPR model requires careful Kernel selection and are sensitive to noise data. A clearer explanation of the distinctions between the models and the criteria for their selection would improve the clarity and justification of the methodology.

2) Line 52-53 outlines the use of RWT and surrogate models to address two key challenges. However, previous studies, for example [1] have already demonstrated that the PCE model can deliver accurate performance in turbine fatigue load prediction, with differences of only around 5% compared to high-fidelity simulations for various site-specific conditions. Furthermore, in this study, 30 seeds are used in simulation, but it remains unclear whether this setup overcomes the computational burden. Since one of the main motivations for surrogate modeling is computational efficiency, this point should be more convincingly justified.

[1] Dimitrov, Nikolay, et al. "From wind to loads: wind turbine site-specific load estimation with surrogate models trained on high-fidelity load databases." *Wind Energy Science* 3.2 (2018): 767-790.

**3-case study:**

- 1) Please clarify 'parameter B away 1' (Line 229).
- 2) Line 240 mentions that simulations were performed to verify the absence of critical resonances under operational scenarios. While this test helps confirm the basic functionality of the turbine model, it does not substitute for validation against real-world turbine model, given that minimal adjustments were made to the RWT model (as stated in lines 238–239). Therefore, a comparison between simulation results and on-site load measurements under similar environmental conditions is recommended. This comparison would help to demonstrate that the turbine model in simulation can predict comparable load as actual turbine.

Meanwhile, this study applies SCADA data to evaluate model accuracy. However, two sources of uncertainty remain: the prediction errors may originate from either the surrogate model or the RWT model used for simulation. As a result, the interpretation of prediction accuracy is unclear.

- 3) In Section 3.4, Line 254 states that 30 seeds were used in the simulation, line 255 explains it as a compromise between small and large sample sizes, but no further justification or explanation is provided for this choice. As discussed in the introduction, the literature review in the manuscript does not establish a clear standard or conclusion for the appropriate number of seeds in such simulations. Given that one of the study's primary motivations is to address computational burden, it is recommended to perform a sensitivity analysis on the number of seeds. This would help assess the impact of wind modeling variability and demonstrate whether using 30 seeds is sufficient to capture the inherent stochasticity of wind conditions, thereby supporting the reliability of the results.
- 4) Line 262 mentions the duration of simulation, but the reason for this specific 630-s is not explained. It is unclear whether 30 s are sufficient to mitigate the initial transient effect, considering the wide speed range. This is important for dynamic responses and load predictions. Further justification, such as signal stabilization plots or references, are suggested to demonstrate.

**4-Results**

 Table 3 compares the prediction errors between the PCE and GPR surrogate models. As the mean MAE and RMSE for mean blade moment and std of tower acceleration are within 10 %, but their std are much enlarged. Line 300 states that the response surfaces of the std are less smooth, resulting in higher errors for these values. However, the reasons for the coarser surfaces are not discussed. More clarifications are suggested.

Also, the influence of model parameters, such as the choice of polynomial order in PCE or kernel type and hyperparameters in GPR, on the accuracy should be addressed. A sensitivity study would help clarify the source of the observed discrepancies. Given the GPR model, which is free from seed number effect, provides similar or slightly lower std values for blade moment. Simply suggesting that a larger seed number would potentially improve smoothness (line 300), is not a sufficient justification without supporting evidence.

This concern also applies to the statement regarding the maximum values due to small parameter variations, where increased seed number may not improve the prediction accuracy. Such claims should be supported by quantitative analysis or validation to avoid speculative conclusions.

Figure 3 compares measured data, openfast simulation results, and predictions by surrogate models. However, the markers appear densely clustered in each subplot, making it a bit difficult to clearly distinguish among different categories.
Moreover, the importance of seed uncertainty mentioned in line 332 is not fully reflected or supported by the figure, as the simulation model differs from the real turbine used in

measurements, and this impact of model uncertainty is not clearly quantified. This uncertainty further limits the evaluation of seed effect.

To improve clarity, it is recommended to separate the comparison into two figures: one comparing measured data with surrogate model predictions, and another comparing measured data with OpenFAST simulation results. This separation would provide a clearer overview of the accuracy, and reliability of both the simulation and surrogate models.

3) Line 355 states that the annual prediction errors could potentially reflect aging trends of the turbine. However, this statement is potentially misleading, as there is no justification provided to confirm that environmental conditions, such as wind speed and turbulence intensity, were comparable between 2017 and 2022. Without such validation, it is difficult to attribute changes in prediction error solely to turbine aging.

A similar concern applies to the statement in Line 360 regarding the role of the pitch controller in extending turbine lifetime. To strengthen these claims, it is recommended to include a comparison of key environmental parameters (e.g., turbulence intensity, wind speed distribution) across the five-year period. In addition, prediction errors should exclude the model uncertainty, as discussed. This would help isolate the influence of external factors, and focus on the ageing impact.

**5-Discussion**

1) The statement regarding the effectiveness and efficiency of the surrogate models in Line 375 is not fully supported by the results presented. While Table 3 shows that the trained PCE and GPR models achieve prediction errors below 10% for the mean blade root moment and the standard deviation of tower acceleration, the errors exceed 10% and even 20%, for the standard deviation of blade root moment and mean tower acceleration, respectively. These discrepancies suggest that the surrogate models seem less reliable for capturing variability than central tendencies.

Furthermore, the claim of efficiency, based on the use of 30 seeds, is not explicitly validated. No sensitivity analysis is provided to assess how the number of seeds influences prediction accuracy, making it difficult to conclude that the chosen setup is computationally efficient or statistically sufficient.

- 2) The uncertainties arising from both the wind turbine model used in OpenFAST and the surrogate models are not independently identified or quantified. Without a clear separation of these sources, it is difficult to determine the contribution of each to the overall prediction error. Therefore, the claim in Line 380 lacks sufficient support. A more rigorous uncertainty analysis, distinguishing between model structural errors and surrogate approximation errors, would be necessary to justify it.
- 3) The limitation mentioned in Line 389-390, regarding model controller actions or system constraints, is not clearly demonstrated or reflected in the presented results. This claim

appears disconnected from the analysis or result, and would benefit from further clarification.

4) As state in lines 394–399, the simplified process limits the generalizability of this study. However, given the prediction errors observed between the surrogate models and the simulation model, it is unconvincing to conclude that simplification alone has the greatest impact on prediction accuracy compared to measurement data. Other factors, such as model assumptions, data quality, and inherent uncertainties, should also be considered and discussed to provide a more comprehensive understanding of the sources of error.

---

## Author Comment (AC1)

Answers to the comments of the reviewer 1 for the manuscript Load Estimation in Onshore Wind Farms Using Surrogate Modeling and Generic Turbine Models submitted to Wind Energy Science

Dear Prof. Vidal,

The authors would like to thank the reviewer for the detailed comments and suggestions to improve the manuscript. In the following, the comments are provided in italic letters. Each point is answered in non-italic letter. Changes to the text of the manuscript are reported in red.

With kind regards,

Alexander Mönnig, Ansgar Hahn, Astrid Lampert and Ulrich Römer

**Reviewer 1**

**General Comments**

**Comment.**

This paper investigates the use of generic turbine models combined with surrogate models to predict loads on real-world turbines, validated through comparison with multi-year measurements from an operational wind farm. The motivation of the paper is highly relevant, addressing a topic of growing interest in the wind energy community. The manuscript is overall well structured and generally well written.

**Answer:** We would like to thank the reviewer for this positive judgment.

**Comment:**

However, the current performance of the surrogate models does not appear sufficient to draw robust conclusions about the method's applicability. I recommend the authors either reframe the manuscript as a methodological paper or extend the work on the surrogate models to improve their predictive capabilities. Specific suggestions are provided below.

**Answer:** To underline the scope of the manuscript we have modified formulations in the abstract and the introduction, to emphasize that the paper primarily investigates the feasibility of using surrogates in connection with generic wind turbine models.

**General Writing**

**Comment:**

Throughout the paper, please integrate references more smoothly into the text rather than relying on "see" or "cf.".

**Answer:** We appreciate this comment and have revised the manuscript with more natural formulations.

**Abstract**

**Comment:**

It is a great short and direct abstract. However, it would benefit from more emphasis on the key challenge the authors are trying to tackle here: the lack of access to commercial turbine models and whether generic models can address high-level design considerations.

**Answer:** Thank you for pointing this out. In the revision we have emphasized this aspect:

This article investigates the feasibility of developing surrogate models, based on slightly adapted generic turbine models, for predicting loads on real-world wind turbines. The approach aims to reduce reliance on proprietary turbine models, typically unavailable to industry practitioners, and is intended to support high-level assessments and decision-making.

**Introduction**

**Comment:**

The introduction is overall good. For clarity, please address the limited access to accurate turbine models first (before the surrogate modelling), as this is the core novelty of the work.

**Answer:** We agree with this and have implemented this change.

**Comment:**

Please include a brief literature review on the use of generic turbine models in other studies.

**Answer:** As suggested we added a short literature review.

**Section 2: Surrogate Models**

**Comment:**

Line 100: Please elaborate on the methodological extension related to the pitch angle.

**Answer:** We agree that a more detailed explanation would be helpful at this point. The new part reads:

For above-rated conditions, where a change in wind speed no longer results in a change of power, the required derivative can no longer be computed. We therefore chose to adapt the method to use the pitch angle  $\beta$  instead, which in pitch regulated machines continuously increases once rated power is reached. Therefore, it represents a suitable proxy for estimating the wind speeds standard deviation as

$$V_{\text{std}} = \frac{\beta_{\text{std}}}{\left(\frac{d\beta}{dV}\right)_{n} \cdot B}.$$
 (1)

**Comment:**

Line 121: The reference to Dimitrov et al. (2018) is abrupt. Adding more context would make it smoother.

**Answer:** Following the reviewers recommendation we have reformulated the text as:

For the wind speed, a Beta distribution could be employed, as already suggested by Dimitrov et al. (2018) who observed that several of their surrogate input variables that depended on the wind speed showed higher variability at lower wind speeds. More samples are therefore needed in that region, making the beta distribution a suitable choice.

**Comment:**

Line 124: the reference to Mara and Becker should also be introduced with more context.

**Answer:** In this case, the reference to Mara and Becker is included only because their paper provides the formula we apply; we therefore did not expand the context further, but reformulated slightly.

**Comment:**

Line 139: The introduction of the forum is somewhat weird. Maybe the best way to do it is to remove this reference and explain this as the procedure you will follow with some justifications. It might also improve clarity to reorganize this as a dedicated section titled "Adapting the Generic Turbine Models," providing a more detailed explanation of each stage of the parameters updates. This section could be just after the introduction.

**Answer:** Thank you for pointing this out. The reference as been removed and

the procedure is now described in more detail, having added some explanations for the required steps. However, the position of the explanation has been kept to reflect its order in the workflow/method.

**Comment:**

Line 150: Please clarify what error thresholds or maximum acceptable errors define the model applicability.

Answer: Previous studies of aeroelastic simulations with detailed turbine models report mean errors for flapwise bending moments and tower top tilt moments typically in the range of 10-20 %, though in some cases the errors can reach 40 %, depending on the quality of the inflow definition. With this in mind, it appears reasonable to expect the errors with additional structural and aerodynamic uncertainties, at least in the range of 50 %. We would argue that these could be still considered useful for initial load estimations and trend analyses, even though they would clearly not be sufficient for detailed design or certification purposes.1

**Comment:**

Line 152: Mentioning SPCE as a future research direction is a bit early in the paper. Consider moving this to the discussion or conclusion, elaborating on its potential. References to other approaches should either be briefly presented with names and results at some point in this paper or not mentioned at all.

**Answer:** Thank you for the comment, we have remove the phrase and added More advanced surrogate modeling approaches could be pursued. For example, a follow-up study tested stochastic polynomial chaos expansion (SPCE), but it did not provide significant improvements over the approaches presented here. to the conclusion.

**Section 3: Case Study**

**Comment:**

Line 205: Please clarify the limits under which generic turbine models can be considered applicable to real-world scenarios.

**Answer:** Thank you for your comment. This point has been addressed in the discussion above.

**Comment:**

Line 224: citations to be improved.

<sup>1M, M. Pedersen, T. J. Larsen, H. A. Madsen, and G. C. Larsen, "More accurate aeroelastic wind-turbine load simulations using detailed inflow information," Wind Energy Science 4, no. 2 (2019): 303–323, https://doi.org/10.5194/wes-4-303-2019

**Answer:** The citation has been revised.

**Comment:**

Section 3.2: The validation of the generic model is lacking. Before introducing surrogate models, a comparison between generic turbine loads and measurements under different DLCs is essential. If this has been done, please include at least key figures, if not, it represents a crucial missing step.

**Answer:** We included a brief validation of model accuracy in the appendix. However, a detailed validation of the generic turbine model against measurements under different DLCs was not feasible within this work, but is an important step for a future study. The study was intended as a proof of concept to evaluate whether the investigated method is useful and promising.

**Comment:**

Figure 2: This figure is currently not cited in the paper. It would improve the methodology section to reference it when discussing the joint distribution and bounds definition.

**Answer:** We thank the reviewer for pointing this out. A reference to Figure has been added in the Section 3.4.

**Comment:**

Line 258: Indicate the OpenFAST version used

**Answer:** Thank you for pointing this out. We now indicate the version of all the tools mentioned in the manuscript.

**Comment:**

Line 264: Justify the use of 30 seconds as transient time with relevant references.

**Answer:** As suggested we now include a reference to existing literature. The revised text reads: An additional 30 seconds at the beginning are included to account for any initial transient effects, which are subsequently excluded from the analysis, consistent with other studies such as the one by Castorrini et al. (2023).

**Comment:**

Line 277: Explain the motivation for using the Rosenblatt transformation with GPR. Have you tuned the hyperparameters (using grid search, random search, or Bayesian optimization)? This is expected to improve model performance.

**Answer:** The Rosenblatt transformation could be omitted within the GPR approach, at the same time the chosen kernel better fits uncorrelated and rescaled data. The hyperparameters have been tuned via maximum likelihood, as

automatically provided by Scikit-learn.

**Section 4: Results**

**Comment:**

Line 290: The opening sentence is unclear. The sensitivity analysis on polynomial order and training set size you mentioned should be first introduced and then presented, including the maximum training set size used.

Answer: In Section 3.6 the training sets of the convergence analysis are now introduced. The other parameter varied in the study, were already included in the submitted manuscript. Furthermore, we added: Since detailed convergence behavior is not the focus of this study, we only summarize the key findings here; full figures and results can be found in Appendix B. to the introduction of Section 4.1 to emphasize that the training results are not discussed in detail, as they revealed nothing new or remarkable.

**Comment:**

Line 300: The current surrogate models seem not sufficiently capture dynamic and nonlinear effects, you should here refer to the other models you tried, and the database should be extended, 300 samples is very little for 4 input parameters.

**Answer:** This study was intended to be a proof of our concept and the sample size was kept purposefully rather small, due to budget and time constraints. Future investigations will include more samples/more replications to better capture the seed-to-seed uncertainty.

**Comment:**

Line 304: Clarify what the "respective other model" refers to.

**Answer:** We agree that this needed to be rephrased for clarity. It now reads: Furthermore, we found that the models trained to predict the mean and standard deviation of the 10-minute maxima performed notably worse than their counterparts for the 10-minute mean or standard deviation.

**Comment:**

Line 307: Specify which convergence analysis is being referenced.

**Answer:** Hopefully, it is clear now with the change of Section 3.6, where the setting of the convergence analysis is introduced.

**Comment:**

Line 310: Provide the mentioned error curves in the manuscript.

**Answer:** We integrated a convergence analysis for our existing training dataset in the appendix - including the error curves. An extension of the data was not possible unfortunately due to the scope of this study.

**Comment:**

Figure 3: The figure is difficult to interpret. Consider restructuring it, good examples could be Figure 3 and 4 in https://doi.org/10.5194/wes-9-1885-2024, showing training sample positions relative to measurements, followed by load predictions relative to inputs.

**Answer:** We agree with this comment and have split the figure into two separate ones. In addition, we have slightly rephrased a related sentence in the description to improve readability. It now reads: The plot over the turbulence intensity shows that measurements at the lower end of the tower acceleration range are not accompanied by nearby predictions, while at higher turbulence intensities several outliers occur, indicating that the surrogate struggles to capture these conditions accurately.

**Comment:**

In addition to that, it can be seen on the figure that many training points do not align with measurements, potentially explaining low model performance.

Answer: We hope that separating the plots improves clarity. When looking at the training data, it indeed appears that many samples lie outside the turbine's typical operating region, which may explain part of the observed deviations. However, when comparing the surrogate predictions with the measurements, the agreement is much closer, indicating that the models capture the turbine's behavior well despite these differences. However, this also indicates that the sampling procedure could potentially be made more efficient, as some of the sampled conditions in the training dataset occur only rarely—either because we are focusing on this specific turbine and/or because only a very small subset of SCADA samples is shown here. This aspect would, however, require closer investigation.

**Comment:**

In Section 4.1, the surrogate models predict mean values adequately (except for maximum tower-top acceleration), but their variance predictions are off. Before moving on to further analysis, the following points need to be studied:

**Answer:** We appreciate this overall remark and address the reviewer's points individually below.

**Comment:**

Quantify the error between generic model outputs under measured environmental conditions and actual measurements.

**Answer:** Regarding the assessment of the model error of the generic turbine

model, we have now added additional data to the Appendix.

**Comment:**

Increase the training set size, 300 samples with four input variables is small. Please show the convergence analysis mentioned earlier and consider extending to 1000 samples for instance.

**Answer:** We have added the convergence analysis to our appendix. However, while we agree that 1000 samples could support our study, it was not feasible to extend the dataset in the scope of this study.

**Comment:**

Perform a convergence study on the number of seeds to take into account to capture the seed-to-seed uncertainty.

**Answer:** We agree that a convergence study on the number of seeds would provide further insight into capturing seed-to-seed uncertainty. Since a convergence study on the number of seeds is beyond the scope of this work, we now list it as a direction for future research in the conclusions.

**Comment:**

Discuss other surrogate models, such as neural networks, which have shown promising results in similar contexts.

**Answer:** This relates back to our limitations in the sample size for this initial feasibility assessment. We believe that there are not enough samples for a NN to be trained, which is why we chose GPR and PCE.

**Comment:**

Address the poor performance at low wind speeds, and elaborate on way to address it, expanding the dataset could be a way.

**Answer:** We are unfortunately not entirely clear on the focus of the question and would kindly ask the reviewer to provide us with some guidance.

**Comment:**

Tune the hyperparameters of the models using grid search or random search for instance.

**Answer:** For the PCE models, the relevant parameters are the polynomial order and regression algorithm, both of which we varied. For the GPR models, scikit-learn tunes the kernel hyperparameters automatically via marginal likelihood optimization, which should serve a similar purpose to something like a grid search. What could have been done for GPR is the test of multiple kernels. But RBF should gave use a decent default behavior, deemed to be good enough for this feasibility assessment. Overall, the focus of the work was not on generating

surrogate models and therefore, we preferred to use well-established standard methods for surrogate modeling.

**Section 5: Discussion**

**Comment:**

Please revise conclusions to reflect that surrogate models captured mean values adequately but struggled with variance and extreme events.

**Answer:** We agree that the conclusions needed better reflect the results of our study.

The updated text is now: Aligning with existing literature, both applied methods, PCE and GPR, were effective in capturing the mean responses. However, predictions of the standard deviations were consistently less accurate than for the means, and this issue was particularly pronounced for the targets involving maximum values. In the case of tower acceleration maxima, no satisfactory surrogate model could be obtained. In general, the PCE approach was found to be more suitable for modeling the blade loads, while GPR was more suitable for predicting the tower acceleration.

**Comment:**

Line 376: The expectation that increasing the number of turbulent seeds will resolve discrepancies lacks evidence. Please conduct and present a convergence analysis on maximum tower acceleration with respect to turbulent seed number and training set size.

**Answer:** In the scope of this study we were not able to implement such an analysis. However, to acknowledge this valid point we aimed to provide a more transparent take on the observed discrepancies.

The modified passage now states: Concerning the developed surrogate models, it can be noted that both the PCE and GPR proved to be effective and efficient at capturing turbine responses of the training data. The challenges in fitting the standard deviation models may be related to the noisy response surfaces. This, however, needs to be investigated in future studies, for instance through additional turbulent realizations and dedicated convergence analyses. Given the lower than anticipated costs for generating the current set of training data, such extensions would be realistic. Therefore, the development of surrogates as a predictive tool remains feasible for industry practitioners, and the current models provide a good baseline for more complex models that may be tested in the future.

**Summary and Recommendations**

**Comment:**

The scope of work is highly relevant, and the preliminary findings on turbine aging and inter-turbine variability are promising.

**Answer:** Again we acknowledge that the reviewer sees the potential in the data and analyses.

**Comment:**

However, substantial improvements are required before publication. Key recommendations are:

**Answer:** We appreciate the reviewer's overall summary and address the individual points below.

**Comment:**

 $Provide\ an\ error\ estimation\ from\ discrepancies\ between\ the\ generic\ OpenFAST\ model\ and\ measurements.$

**Answer:** We have integrated the results of our initial model validation approach, in which some selected SCADA points where compared to OpenFAST model results.

**Comment:**

See the impact of increasing the size of the training dataset on the model performances.

**Answer:** We integrated a convergence analysis for our existing training dataset in the appendix. An extension of the data set was not possible due to the scope of this study.

**Comment:**

Elaborate on surrogate model hyperparameter tuning, considering grid search or random search.

**Answer:** This point has been addressed during the discussion above.

**Comment:**

Expand the discussion on alternative models tested, consider implementing neural networks with larger datasets.

**Answer:** We acknowledge that the surrogate models used in this work are simple and limited, which we consider acceptable given that this is not the main focus of the study. We agree though that more advanced surrogate modeling approaches could be explored, and we have added a brief note mentioning a follow-up study using stochastic polynomial chaos expansion (SPCE). Neural networks were not considered due to the limited amount of available training

data.

**Comment:**

Add sensitivity analyses on the number of turbulent seeds and the size of the training/validation sets.

**Answer:** While we agree that a sensitivity study would be valuable to understand the requirements for modeling the seed-to-seed uncertainty it is beyond the scope of this initial feasibility assessment. We see this as a first step for future work and have clarified this by mentioning the sensitivity study as a follow up topic in the conclusions.

---

## Author Comment (AC2)

Answers to the comments of the reviewer 2 for the manuscript Load Estimation in Onshore Wind Farms Using Surrogate Modeling and Generic Turbine Models submitted to Wind Energy Science

Dear Prof. Vidal,

The authors would like to thank the reviewer for the detailed comments and suggestions to improve the manuscript. In the following, the comments are provided in italic letters. Each point is answered in non-italic letter. Changes to the text of the manuscript are reported in red.

With kind regards,

Alexander Mönnig, Ansgar Hahn, Astrid Lampert and Ulrich Römer

**Reviewer 2**

**General Comments**

**Comment:**

In this study, the authors developed and trained two surrogate models for predicting wind turbine loads and evaluated their performance using SCADA data. The surrogate models exhibited varying levels of accuracy in load prediction. The topic of using surrogate models for turbine performance evaluation is current and relevant to ongoing research in the field. The paper is generally well-structured, with a logical and coherent flow.

**Answer:** We would like to thank the reviewer for this positive judgement.

**Comment:**

However, it does not clearly emphasize its innovations, and the interpretation of the results is not fully convincing. The methodology, including the use of PCE and GPR models and reference wind turbine model, is well-established in the literature, and thus do not present novelty to the reader.

**Answer:** As suggested by Reviewer 1, we have revised the introduction by

moving the discussion of generic surrogate models earlier in the text. This makes our contribution and innovation clearer in the context of established methodology. We agree, that the surrogates used are rather standard. The paper did not aim to introduce novel methods for surrogate modeling, but to investigate the use and potential of generic wind turbine models in combination with surrogates.

We have rephrased parts of the discussion and conclusions to provide a clearer interpretation of the model performance. The revised passages read as follows:

Discussion: For the blade loads, the median prediction errors were consistently within  $10{\text -}15\%$ , a level that can already be considered useful for practical applications. For the tower accelerations, the errors were notably larger, typically around  $30{\text -}40\%$ , yet still within a range that allows the models to support initial load assessments and comparative studies. At the same time, these deviations highlight that further refinement is needed before the surrogates can be used for applications requiring high quantitative accuracy.

Conclusions: For the blade loads, the surrogates achieved median relative prediction errors of around 10% for maximum loads and 10–20% for the mean loads. This accuracy suggests that the blade load models can already be applied for practical purposes such as initial load assessments. In contrast, the tower acceleration models performed notably worse, with median prediction errors typically between 30–40%, and even higher errors for maximum accelerations, limiting their current usability.

**Comment:**

Furthermore, the interpretation of the prediction accuracy is questionable. For example, the annual median error in tower acceleration exceeds 40%, which raises concerns not only about the accuracy of the surrogate model itself, but also about the consistency between the reference turbine used in the OpenFAST simulation and the actual on-site load measurements. This concern hinders the reproduction, generalization, and practical application of the proposed approach (surrogate + generic WT models).

Answer: Indeed, the simplifying assumptions in the simulation of the reference turbine do contribute to the deviation between the surrogate and the SCADA data. We are now reporting simulation results comparing both for selected operating conditions, quantifying roughly the order of the modeling error. The surrogate error contributes further and is now also investigated in more detail in the appendix. At the same time, even in view of the deficiencies of the surrogate and reference turbine modeling approach, errors of acceptable size for practical applications could be obtained. The approach is clearly described and reproducible, in particular with the published code at the same time we agree that the aspect of generalization requires more research with larger data sets.

The attached comments are provided for each section of the manuscript, with the hope of improving the quality of the paper.

**Answer:** We are grateful for the detailed comments and improved the manuscript based on the suggestions and questions. Below, each point is addressed, including the quotation of changes done to the manuscript.

**Abstract**

**Comment:**

The authors present a concise abstract that briefly introduces the methodologies applied in the study, and the conclusion regarding the use of surrogate models for load prediction. However, the research question is not clearly stated, leaving the reader uncertain about the specific problem the study aims to address.

**Answer:** This point was also made by reviewer 1. We agree with this assessment and have revised the abstract accordingly to explicitly state the problem our study addresses.

**Comment:**

In addition, the claim of achieving 'reasonable accuracy' (line 7) requires clarification, as the blade load and tower acceleration report distinguish errors. This variation should be acknowledged and discussed more explicitly.

**Answer:** We agree and rephrased the sentence to: While the models approximated the real-world turbine behavior with reasonable accuracy for the blade loads, the prediction quality for tower accelerations was notably lower.

**Introduction**

**Comment:**

In this section, the authors identify two key challenges in aerodynamic simulations: high computational cost and limited access to detailed turbine models. Surrogate models are then proposed as a means to reduce computational effort, while the adoption of a reference wind turbine model is suggested as a potential solution to the issue of limited public model. However, the motivation or assumptions the selection of PCE and GPR models require further clarification. In addition, the literature review does not identify existing gaps in the application of surrogate models, particularly PCE or GPR, for wind turbine load prediction. As a result, the novelty or contribution of the study is not convincingly demonstrated.

**Answer:** We agree with the point, that the novelty of the research needed highlighting. This was also mentioned by reviewer 1 and has been addressed

above. The novelty of the work is also now stated directly in the introdcution: While surrogate modeling is therefore an established approach, the novelty of this study lies in assessing the feasibility of combining it with generic reference turbine models to predict real turbine loads on a large scale, thereby reducing reliance on commercial models that are typically inaccessible. To this end, we use a database of five years of SCADA data to evaluate the accuracy of the combined surrogate–RWT approach.

Moreover, we aimed to clarify the choice for PCE and GPR in the introduction, by rephrasing the relevant passage to: Given that the turbulence-induced variance sets a natural limit on predictive accuracy, our surrogate models - based on PCE and GPR - will estimate both the mean and the standard deviation of the loads, as these approaches have been proven effective in similar applications in the literature.

**Comment:**

Although the authors aim to address data limitations by using a reference turbine model, the lack of either public available real turbine models or SCADA datasets makes it difficult to assess the broader significance of this study or novelty of the approach.

Answer: We agree that the lack of publicly available turbine models and SCADA datasets is a challenge for the community. The absence of real turbine models was the main motivation for our study. Furthermore, the SCADA dataset used here has been published, so our case study can be reproduced and built upon. It is correct, however, that extending the approach to other turbine types would require the publication of additional datasets, which could be facilitated by future studies.

**Comment:**

1) The authors state that the development of surrogate models remains costintensive due to the randomness associated with wind turbulence. However, the discussion around the number of seeds used in simulations is unclear. While the cited literature uses varying seed numbers (e.g., 6, 100, 8), the paper does not provide a clear conclusion on what constitutes a sufficient number of seeds for accurate load prediction. It would be valuable to clarify how the number of seeds affects the reliability or prediction accuracy of the surrogate model, and what the potential implications are of using an insufficient number, such as the recommended value of 6 by IEC standard.

Answer: In this study, we aimed to simulate as many seeds as possible, but budget and time constraints limited us to 30 realizations. Since a systematic convergence analysis on the required number of seeds requires additional data, we now state this as a point for additional research in a follow up study. Still, the effect of the limited sample size is roughly visible in Figure 4, where standard deviations due to the seed-to-seed variability are drawn; indicating a manageable effect for the blade-related quantities and a large relevance for the

tower-related outputs.

**Comment:**

While the PCE model used in Murica et al. (2018) required 100 realizations to predict the mean and standard deviation of the load, the authors of this study describe that approach as having a "high computational cost" (line 40). However, this study also adopts the PCE model and claims it "can overcome the computational burden" (line 40), without providing a clear explanation of the number of seeds used or the rationale for selecting the PCE model over other alternatives such as ANN, RSM, or GPR, which are mentioned briefly in lines 25–26. A more detailed justification for the model choice and a discussion of its computational trade-offs would strengthen the study, and reduce potential confusion for readers.

Answer: Thank you for pointing this out. The statement of high computational cost is indeed misleading in this context as formulated and has been removed. Ideally, one should formulate a target accuracy (e.g. for the mean response) and carry out as many seed-to-seed replications as needed. As argued above, due to budget and time constraints we were not able to evaluate the required size precisely. However, no matter how large the required sample size, the PCE approach is expected to provide a much faster access to this mean value over the entire parameter space, compared to purely evaluating the simulation model and this is what is meant by can overcome the computational burden. We didn't pursue NNs because of the small sample size, which is now mentioned in the text and chose GPR and PCE because of their known ability to work in the small-to-moderate data regime, with a moderate number of inputs. We added "Moreover, PCE and GPR were chosen as relatively simple methods that perform well even with limited training data, making them well-suited for this initial feasibility study." to the introduction.

**Comment:**

In line 44, the GPR model, described as not relying on replication, is also selected in this study, which helps to overcome the high computational burden associated with turbulence-induced randomness. However, the differences between the PCE and GPR models are not clearly presented, making the rationale behind the selection and comparison of these two models unclear to the reader. Especially, the GPR model requires careful Kernel selection and are sensitive to noise data. A clearer explanation of the distinctions between the models and the criteria for their selection would improve the clarity and justification of the methodology.

**Answer:** Here the GPR model was applied in the same way, as a pure regressor to approximate mean and standard deviation separately. Hence, we didn't study heteroscedastic GPs that could indeed work without replications. Hence, there is not much difference in the way the methods are applied and we decided therefore, to keep these parts short. Indeed, choosing a kernel is relevant and difficult but the chosen one is widely used and appeared to be a reasonable

choice for these first attempts.

**Comment:**

Line 52-53 outlines the use of RWT and surrogate models to address two key challenges. However, previous studies, for example [1] have already demonstrated that the PCE model can deliver accurate performance in turbine fatigue load prediction, with differences of only around 5% compared to high-fidelity simulations for various site-specific conditions. Furthermore, in this study, 30 seeds are used in simulation, but it remains unclear whether this setup overcomes the computational burden. Since one of the main motivations for surrogate modeling is computational efficiency, this point should be more convincingly justified. [1] Dimitrov, Nikolay, et al. "From wind to loads: wind turbine site-specific load estimation with surrogate models trained on high-fidelity load databases." Wind Energy Science 3.2 (2018): 767-790.

Answer: The main computational savings of our method, come from the fact that once the surrogate models are trained, they can be used to evaluate loads across the full SCADA dataset without the need for costly aeroelastic simulations. In this sense, the setup overcomes the computational burden mentioned. We have also added this explanation into our introduction to improve clarity. As mentioned above, the seed size should not be chosen ad-hoc but determined based on an estimated of the associated sampling error. This was unfortunately not feasible in our study due to the aforementioned constraints.

**Section 3: Case Study**

**Comment:**

Please clarify 'parameter B away 1' (Line 229).

**Answer:** Thank you for pointing this out. We changed this to: However, due to the absence of mast wind measurements, there is no empirical basis to calibrate the parameter B. Therefore, it is kept at a value of 1 in this study.

**Comment:**

Line 240 mentions that simulations were performed to verify the absence of critical resonances under operational scenarios. While this test helps confirm the basic functionality of the turbine model, it does not substitute for validation against real-world turbine model, given that minimal adjustments were made to the RWT model (as stated in lines 238–239). Therefore, a comparison between simulation results and on-site load measurements under similar environmental conditions is recommended. This comparison would help to demonstrate that the turbine model in simulation can predict comparable load as actual turbine.

**Answer:** We have integrated the results of our initial model validation approach, in which some selected SCADA points where compared to OpenFAST

model results, with the aim of estimating the deviation between the adapted reference turbine model and our on site measurements.

**Comment:**

Meanwhile, this study applies SCADA data to evaluate model accuracy. However, two sources of uncertainty remain: the prediction errors may originate from either the surrogate model or the RWT model used for simulation. As a result, the interpretation of prediction accuracy is unclear.

Answer: It is correct that we are dealing with different sources for uncertainty, which are contributing to the observed prediction errors: the surrogate models themselves and the underlying generic turbine model used in the simulations. Given that the applied surrogates showed good fits for the mean responses (except for the towers maximum acceleration) - while larger discrepancies only appeared in the case study - it is likely that the dominant contribution to the overall error stems from the generic turbine representation and the limited set of input parameters considered. However, while a quantitative separation of these contributions is difficult and remains an open task for future work, we integrated an overview of uncertainties in the modeling workflow to increase transparency.

**Comment:**

In Section 3.4, Line 254 states that 30 seeds were used in the simulation, line 255 explains it as a compromise between small and large sample sizes, but no further justification or explanation is provided for this choice. As discussed in the introduction, the literature review in the manuscript does not establish a clear standard or conclusion for the appropriate number of seeds in such simulations. Given that one of the study's primary motivations is to address computational burden, it is recommended to perform a sensitivity analysis on the number of seeds. This would help assess the impact of wind modeling variability and demonstrate whether using 30 seeds is sufficient to capture the inherent stochasticity of wind conditions, thereby supporting the reliability of the results.

Answer: As pointed out above, the number of seeds does not aim to reduce the computational burden of our proposed method, since the main savings are obtained in the application of the surrogates. We agree however, that a sensitivity study on the number of seeds, should be conducted in a follow up project. To clarify, we revised the sentence concerning the choice for the number of seeds: This seed number aligns with TurbSim's documentation [1] as a compromise between the small and large sample sizes found in the literature. While a larger number of seeds would be desirable, with the required number depending on the desired statistical accuracy, the choice of 30 represents a practical limit for this first feasibility study.

4) Line 262 mentions the duration of simulation, but the reason for this specific 630-s is not explained. It is unclear whether 30 s are sufficient to mitigate the initial transient effect, considering the wide speed range. This is important for dynamic responses and load predictions. Further justification, such as signal stabilization plots or references, are suggested to demonstrate.

**Answer:** This was also pointed out by the reviewer 1 and references are now provided.

**Section 4: Results**

**Comment:**

Table 3 compares the prediction errors between the PCE and GPR surrogate models. As the mean MAE and RMSE for mean blade moment and std of tower acceleration are within 10%, but their std are much enlarged. Line 300 states that the response surfaces of the std are less smooth, resulting in higher errors for these values. However, the reasons for the coarser surfaces are not discussed. More clarifications are suggested.

**Answer:** We agree with this comment and have expanded the explanation in Section 4.1.1 to clarify the potential reasons for the less smooth response surfaces. The revised text reads:

This lack of smoothness may arise because the estimates of the standard deviations have not yet converged with only 30 turbulent realizations, or because the underlying surfaces are inherently less smooth. Using a larger dataset with more realizations could help clarify this and potentially reduce the observed errors.

**Comment:**

Also, the influence of model parameters, such as the choice of polynomial order in PCE or kernel type and hyperparameters in GPR, on the accuracy should be addressed. A sensitivity study would help clarify the source of the observed discrepancies. Given the GPR model, which is free from seed number effect, provides similar or slightly lower std values for blade moment. Simply suggesting that a larger seed number would potentially improve smoothness (line 300), is not a sufficient justification without supporting evidence.

Answer: We would like to clarify that both the PCE and GPR models rely on the 30 turbulent realization in order to estimate the mean and standard deviation of the target responses. The impact of polynomial order and regressor is addressed in the presentation of the training results. However, we did not conduct a sensitivity study on the kernel choice for GPR. Fitting a Matern kernel would be a logical next step, including also sample paths with reduced regularity. At the same time, we expect that improving the model itself promises more gains in accuracy towards the recorded data. An analysis of the convergence

behavior of our models is now integrated in the appendix.

**Comment:**

This concern also applies to the statement regarding the maximum values due to small parameter variations, where increased seed number may not improve the prediction accuracy. Such claims should be supported by quantitative analysis or validation to avoid speculative conclusions.

**Answer:** We agree and have now removed this speculative statement.

**Comment:**

Figure 3 compares measured data, openfast simulation results, and predictions by surrogate models. However, the markers appear densely clustered in each subplot, making it a bit difficult to clearly distinguish among different categories.

**Answer:** The figure has been reworked by splitting it into two subfigures and reducing the number of plotted points.

**Comment:**

Moreover, the importance of seed uncertainty mentioned in line 332 is not fully reflected or supported by the figure, as the simulation model differs from the real turbine used in measurements, and this impact of model uncertainty is not clearly quantified. This uncertainty further limits the evaluation of seed effect.

**Answer:** The importance of seed-to-seed uncertainty is derived directly from the 30 replicated simulations, as reflected by the error bars in the figure. While the absolute magnitude may differ for the real turbine, we expect the present and qualitative relevance of this uncertainty to transfer to the real turbine.

**Comment:**

To improve clarity, it is recommended to separate the comparison into two figures: one comparing measured data with surrogate model predictions, and another comparing measured data with OpenFAST simulation results. This separation would provide a clearer overview of the accuracy, and reliability of both the simulation and surrogate models.

**Answer:** To improve clarity, we have separated the results into two figure and reduced the amount of sampled data so the plots are less crowded.

Line 355 states that the annual prediction errors could potentially reflect aging trends of the turbine. However, this statement is potentially misleading, as there is no justification provided to confirm that environmental conditions, such as wind speed and turbulence intensity, were comparable between 2017 and 2022. Without such validation, it is difficult to attribute changes in prediction error solely to turbine aging.

**Answer:** As recommended, we added two figures illustrating the annual distributions of wind speed and turbulence intensity and have refined the accompanying analysis.

**Comment:**

A similar concern applies to the statement in Line 360 regarding the role of the pitch controller in extending turbine lifetime. To strengthen these claims, it is recommended to include a comparison of key environmental parameters (e.g., turbulence intensity, wind speed distribution) across the five-year period. In addition, prediction errors should exclude the model uncertainty, as discussed. This would help isolate the influence of external factors, and focus on the ageing impact.

**Answer:** We agree that a separation of model uncertainty and external influences would be useful. However, given the scope of this feasibility study, it was not possible to isolate model uncertainty quantitatively. The concern about a comparison of the mentioned parameter has been addressed above.

**Section 5: Discussion**

**Comment:**

The statement regarding the effectiveness and efficiency of the surrogate models in Line 375 is not fully supported by the results presented. While Table 3 shows that the trained PCE and GPR models achieve prediction errors below 10% for the mean blade root moment and the standard deviation of tower acceleration, the errors exceed 10% and even 20%, for the standard deviation of blade root moment and mean tower acceleration, respectively. These discrepancies suggest that the surrogate models seem less reliable for capturing variability than central tendencies.

Answer: We now hightlight this observeration more directly in our manuscript: Concerning the developed surrogate models, it can be noted that both the PCE and GPR were generally effective and efficient at capturing the mean responses of the training data, but were less successful for the standard deviations, and no satisfactory fits could be obtained for the maximum tower accelerations.

Furthermore, the claim of efficiency, based on the use of 30 seeds, is not explicitly validated. No sensitivity analysis is provided to assess how the number of seeds influences prediction accuracy, making it difficult to conclude that the chosen setup is computationally efficient or statistically sufficient.

**Answer:** As discussed above the choice of 30 turbulent seeds does not aim to reduce computational burden. We agree that it represent a limiting factor in how accurately we can capture the seed to seed uncertainty. Concerning the computational efficiency, we consider explicit validation unnecessary, since applying surrogates to the full SCADA dataset is by design far more efficient than simulating every operating point with the aeroelastic code.

**Comment:**

The uncertainties arising from both the wind turbine model used in OpenFAST and the surrogate models are not independently identified or quantified. Without a clear separation of these sources, it is difficult to determine the contribution of each to the overall prediction error. Therefore, the claim in Line 380 lacks sufficient support. A more rigorous uncertainty analysis, distinguishing between model structural errors and surrogate approximation errors, would be necessary to justify it.

Answer: To clarify, at this point in the text, our statement does not refer to predictive accuracy but to the practical feasibility of developing surrogate models in an industrial context. Later on in the discussion we talk about how usable the approach seems, with the observed prediction errors in mind. Concerning the separation of uncertainties, we agree that it would be a valuable next step. However, we are of the opinion that, when comparing the relatively low surrogate validation errors (of the mean responses) with the larger prediction errors observed in the case study, most of the discrepancy is likely attributable to the turbine model and its simulation rather than the surrogates themselves.

**Comment:**

The limitation mentioned in Line 389-390, regarding model controller actions or system constraints, is not clearly demonstrated or reflected in the presented results. This claim appears disconnected from the analysis or result, and would benefit from further clarification.

**Answer:** We agree with this comment and have revised the sentence: This indicates that the approach cannot fully account for changing operating conditions, whether driven by environmental factors or technical modifications, due to modeling constraints and the limited availability of detailed information.

As state in lines 394–399, the simplified process limits the generalizability of this study. However, given the prediction errors observed between the surrogate models and the simulation model, it is unconvincing to conclude that simplification alone has the greatest impact on prediction accuracy compared to measurement data. Other factors, such as model assumptions, data quality, and inherent uncertainties, should also be considered and discussed to provide a more comprehensive understanding of the sources of error.

Answer: We agree with this assessment and have adjusted the manuscript to be more balanced: This simplification is expected to have had a major impact on the prediction accuracy achieved in the presented case study. However, other factors, such as the limitation to four input features, their calculation (e.g. in the case of the air density), the relatively coarse grid for the simulation of the wind field or data quality issues in the SCADA dataset, also need to be considered as contributing sources. A detailed overview of expected uncertainties in our modeling workflow is presented in the Appendix.